# In Vitro Antileishmanial and Antitrypanosomal Activities of Plicataloside Isolated from the Leaf Latex of *Aloe rugosifolia* Gilbert & Sebsebe (Asphodelaceae)

**DOI:** 10.3390/molecules27041400

**Published:** 2022-02-18

**Authors:** Gete Chemeda, Daniel Bisrat, Mariamawit Y. Yeshak, Kaleab Asres

**Affiliations:** 1Department of Pharmaceutical Chemistry and Pharmacognosy, School of Pharmacy, College of Health Sciences, Addis Ababa University, Addis Ababa P.O. Box 1176, Ethiopia; gete.chemeda@ju.edu.et (G.C.); daniel.bisrat@aau.edu.et (D.B.); mariamawit.yonathan@aau.edu.et (M.Y.Y.); 2School of Pharmacy, Faculty of Health Sciences Institute of Health, Jimma University, Jimma P.O. Box 378, Ethiopia

**Keywords:** antitrypanosomal, antileishmanial, *Aloe rugosifolia*, *Trypanosoma congolense*, *Leishmania aethiopica*, *Leishmania donovani*, plicataloside, Asphodelaceae

## Abstract

Trypanosomiasis and leishmaniasis are among the major neglected diseases that affect poor people, mainly in developing countries. In Ethiopia, the latex of *Aloe rugosifolia* Gilbert & Sebsebe is traditionally used for the treatment of protozoal diseases, among others. In this study, the in vitro antitrypanosomal activity of the leaf latex of *A. rugosifolia* was evaluated against *Trypanosoma congolense* field isolate using in vitro motility and in vivo infectivity tests. The latex was also tested against the promastigotes of *Leishmania aethiopica* and *L. donovani* clinical isolates using alamar blue assay. Preparative thin-layer chromatography of the latex afforded a naphthalene derivative identified as plicataloside (2,8-*O*,*O*-di-(β-D-glucopyranosyl)-1,2,8-trihydroxy-3-methyl-naphthalene) by means of spectroscopic techniques (HRESI-MS, ^1^H, ^13^C-NMR). Results of the study demonstrated that at 4.0 mg/mL concentration plicataloside arrested mobility of trypanosomes within 30 min of incubation period. Furthermore, plicataloside completely eliminated subsequent infectivity in mice for 30 days at concentrations of 4.0 and 2.0 mg/mL. Plicataloside also displayed antileishmanial activity against the promastigotes of *L. aethopica* and *L. donovani* with IC_50_ values 14.22 ± 0.41 µg/mL (27.66 ± 0.80 µM) and 18.86 ± 0.03 µg/mL (36.69 ± 0.06 µM), respectively. Thus, plicataloside may be used as a scaffold for the development of novel drugs effective against trypanosomiasis and leishmaniasis.

## 1. Introduction

Neglected tropical diseases (NTDs) consist of a group of diseases which currently account about 12% of the total global health burden in 149 tropical and subtropical countries [1,2]. Among these, trypanosomiasis and leishmaniasis are the two major NTDs in Africa [3]. Trypanosomiasis is a disease caused by the parasitic protozoa of the genus *Trypanosoma*, which is potentially fatal to both human and animal [4]. This disease mainly occurs in tropical Africa and South America [5]. Human African Trypanosomiasis (HAT), also known as sleeping sickness, is caused by two subspecies of Trypanosoma parasites, *Trypanosoma brucei gambiense* and *Trypanosoma brucei rhodesiense* [6], of which the former accounts for 98% of the cases [7]. There are also reports that indicate that other trypanosome species such as *T. congolense* and *T. evansi* cause human trypanosomiasis [8] in regions ranging from the Sahara to the Kalahari Desert [9]. 

Leishmaniasis is a parasitic disease caused by the protozoan parasite *Leishmania* [10]. The disease is prevalent in Africa, the Mediterranean basin, Asia, Latin America, and the Middle East, and has recently been identified in East Timor, Thailand and in Kangaroos in Australia [11]. In Ethiopia, visceral leishmaniasis occurs mainly in arid and semiarid areas [12], whereas cutaneous leishmaniasis is prevalent in highland areas [13]. Leishmaniasis is associated with malnutrition, population displacement, poor housing, a weak immune system, and a lack of financial resources. Depending upon the forms of the disease, it leads to various health problems. Cutaneous leishmaniasis causes skin lesions that leave life-long scars and serious disability, while mucocutaneous leishmaniasis leads to partial or total destruction of mucous membranes of the nose, mouth, and throat. Visceral leishmaniasis is fetal if left untreated. Similarly, the major symptoms of African trypanosomiasis include fever, skin lesions, and swollen lymph nodes on the back of the neck. The infection may become meningoencephalitis. If the patient fails to receive medical treatment, death will occur, usually within months. 

Recently, there have been some attempts to control protozoal diseases, partly because they have become more difficult to control due to a number of factors that limit the utility of current drugs in resource-poor settings, such as high cost, poor compliance, drug resistance, low efficacy, and poor safety [14,15,16]. Therefore, safe, non-toxic, and cost-effective drugs are urgently required to eliminate this problem from every corner of world [7,17]. The latex of *Aloe rugosifolia* Gilbert & Sebsebe is traditionally used for treatments of malaria, wounds, worms, and internal parasites in human and livestock in Hamer district of southern Ethiopia [18]. Latexes of several *Aloe* species have been reported to possess antiprotozoal activity [19,20,21]. Thus, in this study, the antileishmanial and antitrypanosomal activities of *Aloe rugosifolia* and its major constituent plicataloside have been investigated. 

## 2. Results

### 2.1. Structural Elucidation of Plicataloside

The leaf latex of *A. rugosifolia* showed only one peak when analyzed by HPLC (Appendix A). Following this, application of the leaf latex of *A. rugosifolia* on silica gel- PTLC using chloroform: methanol (2:1) as a solvent system led to the isolation of a light-brown-coloured amorphous substance with *R*_f_ value of 0.27. A pseudomolecular ion at *m*/*z* 513.1622 was observed in the negative-mode HRESI-MS as the deprotonated molecular ion [M−H]^−^ (Appendix A), which confirmed the molecular formula to be C_23_H_30_O_13_ (calcd. *m/z* = 513.1608 [M−H]^−^). Two main fragment ions at *m*/*z* 351.1095 and 189.0565 were also observed in the negative-mode HRESI-MS, corresponding to the loss of a glucose moiety and a subsequent loss of another glucose moiety from the first fragment ion, respectively.

The ^13^C-NMR (Appendix A) and DEPT-135 (Appendix A) spectral data of the compound indicated the presence of 23 carbon atoms. Of these, 17 carbon atoms appeared in the DEPT-135 spectrum, including one sp^3^ methyl (CH_3_), eight sp^3^ oxymethines (OCH), two sp^3^ dioxymethines (OCHO), two sp^3^ oxymethylenes (OCH_2_), and four aromatic sp^2^ methines (=CH). Differential profiling of ^13^C-NMR and DEPT-135 spectral data allowed the identification of six quaternary aromatic carbons (δ 115.21, 133.25, 134.00, 140.04, 144.49, and 154.37) in the compound. Diglucosylation was evident from the ^13^C-NMR spectral data (two anomeric carbons (δ 103.05, 104.58), two sp^3^ oxymethylene carbons (δ 61.18, 61.32) and eight sp^3^ oxymethine carbons (δ 70.29, 70.29, 74.05, 74.66, 76.59, 76.75, 77.47, 77.92).

The ^1^H-NMR spectrum (Appendix A) of the compound (see Materials and Methods section) revealed the presence of four aromatic protons assigned as H-4 (δ 7.19, 1H, *brs*), H-5 (δ 7.40, 1H, *brd,* J = 8 Hz), H-6 (δ 7.26, 1H, *t,* J = 8 Hz), and H-7 (δ 7.19, 1H, *brd,* J = 8 Hz). Diglucosylation was also confirmed by the ^1^H-NMR spectrum as indicated by twelve proton signals between δ 2.51 and 3.44, along with other two anomeric protons as H-1′ (δ 4.82, 1H, *d,* J = 8 Hz) and H-1**^″^** (δ 4.99, 1H, *d*, J = 8 Hz). The positions of attachment of the two *O*-glucosyl units at C-2 and C-8 were confirmed by HMBC experiments (Appendix A), which revealed correlations between the anomeric proton H-1′ (δ 4.82, *d*) and C-2 (δ 140.04) and also between the anomeric proton H-1**^″^** (δ 4.99, *d*) and C-8 (δ 154.37). Two singlets were detected at δ 2.38 and δ 9.30 due to the presence of a methyl group and a hydroxyl group attached to a naphthalene ring, respectively. From the above spectroscopic data and comparison with those reported in the literature [22,23], the compound was identified as 2,8-*O*,*O*-di-(β-D-glucopyranosyl)-1,2,8-trihydroxy-3-methylnaphthalene (plicataloside) (Figure 1). 

### 2.2. Oral Acute Toxicity

Oral administration of plicataloside (2000 mg/kg) led to minor signs of toxicity such as temporary shivering and diarrhea. However, these signs disappeared after 24 h. No mortality was also observed following the administration of plicataloside (2000 mg/kg). This implies that plicataloside is safe up to the dose levels used. As per the OECD guideline, its LD_50_ is above 2 g/kg [24].

### 2.3. In Vitro Antitrypanosomal Activity

The current study demonstrated that the incubation of infected blood with 4 and 2 mg/mL concentrations of both the latex and plicataloside arrested the motility of *T. congolense* within the first hour of incubation period. Similarly, at concentrations of 4.0, 2.0, and 0.4 mg/mL, diminazene diaceturate completely eliminated trypanosomal motility within the first hour (Table 1). The results suggested that that *A. rugosifolia* has promising in vitro antitrypanosomal activity. 

### 2.4. In Vivo Infectivity Test

As shown in Table 2, the leaf latex of *A. rugosifolia* and plicataloside inhibited healthy mice from developing infection for more than 30 days or significantly prolonged the pre-patent period especially at a concentration of 4.0 mg/mL unlike the animals in the negative control group, which developed infection within 11 days after inoculation.

### 2.5. In Vitro Antipromastigote Activity

IC_50_ (effective concentration required to achieve 50% growth inhibition) values of the latex and plicataloside were determined against promastigotes of *L. aethiopica* and *L. donovani* in vitro (Table 3). The current study revealed that the leaf latex of *A. rugosifolia* possesses antileishmanial activity with IC_50_ values of 24.5±0.24 and 31.21±0.01 μg/mL on the promastigotes stage of *L. aethiopica* and *L. donovani*, respectively. On the other hand, plicataloside showed stronger activity when compared to that of the latex against both promastigotes of *L. aethiopica* and *L. donovani* with IC_50_ values of 14.22±0.41 μg/mL (27.66 ± 0.80 µM) and 18.86±0.03 μg/mL (36.69 ± 0.06 µM), respectively. 

## 3. Discussion

Trypanosomiasis and leishmaniasis are the two major neglected tropical diseases (NTDs), which currently affect poor people mainly in developing countries [1,2]. Several plants from the genus *Aloe* are endowed with antiprotozoal activities [19,20,21,25,26,27,28]. As a continuation of our search for antiprotozoal compounds from the genus *Aloe*, the leaf latex of *A. rugosifolia* and its major compound were investigated for their in vitro antitrypanosomal and antileishmanial activities. 

As shown in Table 1, both the leaf latex of *A. rugosifolia* and its major compound plicataloside showed in vitro activity against *T. congolense* parasites. Plicataloside further eliminated subsequent infectivity in mice for 30 days. It has been reported that the motility of parasites constitutes a relatively reliable indicator of viability among most zoo flagellate parasites [29]. In the in vitro system when no extract is present, it has been proved that trypanosomes can survive for about 4 h [30]. Trypanocidal activity was measured based on comparison of drop or cessation in motility between the parasites in test substances treated blood- and parasite-loaded control blood without test substances [30,31]. Generally, the shorter the time of cessation of motility of the parasite, the more active the extract is considered to be [30].

A previous study on the leaf latex of *Aloe gilbertii*, its major compound aloin, and its derivatives demonstrated promising antitrypanosomal effects [25]. The report indicates that at a concentration of 4.0 mg/mL, rhein, aloe-emodin, the latex, and aloin completely immobilized or killed the parasites within 15, 20, 25, and 40 min, respectively. Another study by Tadesse et al. [32] demonstrated that the dichloromethane and methanol leaf extracts of *Dovyalis abyssinica* ceased the motility of *T. congolense* within 1 h at concentrations of 20.0, 10.0, and 2.0 mg/mL. Furthermore, both extracts eliminated subsequent infectivity in mice. Several plant extracts have also been reported to have antitrypanosomal activities against *T. congolense* [30,33,34]. Previous reports [29,30,35] indicate that the mean minimum inhibitory concentration (MIC) value of the common trypanocidal drugs is 10.7 mg/mL, and that agents with MIC values of 5–20 mg/mL could be regarded as very active. Therefore, the leaf latex of *A. rugosifolia* and plicataloside can be considered as promising antitrypanosomal agents, although their activity was not as strong as the positive control diminazene diaceturate. 

It should be noted that in the in vitro model, complete immobility of parasites does not necessarily mean that the parasites are dead, but rather the parasites may have lost their infectivity. This was confirmed through the infectivity test, which showed that the leaf latex of *A. rugosifolia* and plicataloside inhibited healthy mice from developing infection for more than 30 days or significantly prolonged the pre-patent period, especially at a concentration of 4.0 mg/mL (Table 2), unlike the animals in the negative control group, which developed infection within 11 days after inoculation. 

Even though the lowest concentration of both plicataloside and diminazene diaceturate (0.4 mg/mL) highly reduced the motility of the parasite, mice administered with these doses contracted infection. Thus, it appears reasonable to speculate that these doses may belong to the group that acts by static action affecting the growth and multiplication of trypanosomes rather than eliminating them completely [30]. These results are consistent with other findings reported elsewhere [25,32]. 

The latex of *A. rugosifolia* also showed good antileishmanial activity against the promastigotes of *L. aethiopica* (IC_50_ = 24.5 ± 0.24 µg/mL) and *L. donovani* (IC_50_ = 31.21 ± 0.01 µg/mL), although its effect was inferior to that of plicataloside, which showed IC_50_ values of 14.22 ± 0.41 and 18.86 ± 0.03 µg/mL, respectively. These activities are much lower than those reported for the latex of *Aloe macrocarpa* (IC_50_ = 1.90 and 1.92 μg/mL, respectively) [36], but higher than those reported for *Aloe vera* leaf exudate against *L. donovani* strains (IC_50_ = 70–115 μg/mL) [19]).

As shown in Table 3, although the antipromastigote activity of plicataloside was less than that of the positive-control amphotericin B (IC_50_ = 8.1 ± 0.11 µg/mL (8.77 ± 0.12 µM) and 7.20 ± 0.15 µg/mL (7.79 ± 0.16 µM) against *L. aethopica* and *L. donovani*, respectively), it was more potent than the antileishmanial drug pentostam, which showed an IC_50_ value of 28.41 μg/mL against *L. donovani* promastigotes [37]. Plicataloside is a naphthalene derivative first isolated from the leaf latex of South African *Aloe plicatilis* [22] and then from the leaf latex of Ethiopian *Aloe otallensis* [23]. Several biological activities including antimalarial and antioxidant [22] and antityrosinase [38] have been associated with it. 

It is difficult to speculate the mechanism by which these test substances exhibit their trypanocidal action. However, accumulated evidence suggests that many natural products show trypanocidal activity by interfering with the redox balance of the parasites [39]. A number of natural products possess structures which are capable of generating radicals that can cause peroxidative damage to trypanothione reductase which is very sensitive to alterations in redox balance. It is also known that some compounds act by binding with the kinetoplast DNA of the parasite [39].

Perusal of literature unveils that a large number of test substances show relatively higher activities against the promastigotes of *L. aethopica* than those of *L. donovani* [36,40,41]. These differences in drug susceptibility between the two species could be associated with slight differences in their membrane composition and/or metabolic characteristics. The extent to which the two parasites are exposed to different chemical substances and culture conditions may also affect their susceptibility to drugs [42,43]. To the best of our knowledge, this is the first report on antitrypanosomal and antilishmanial activities of the leaf latex of *A. rugosifolia* and its major compound plicataloside. Future optimization of plicataloside through structural alteration may lead to synthesis of molecules with improved trypanocidal and leishmanicidal activities.

## 4. Materials and Methods

### 4.1. Plant Material 

Fresh leaves of *A. rugosifolia* were collected in September 2019 from Dida Chena plain between Yabelo and Mega, Oromia region, southern Ethiopia. The plant was authenticated by Mr. Anteneh Desta, the National Herbarium, Department of Plant Biology and Biodiversity Management, College of Natural and Computational Sciences, Addis Ababa University (AAU), where a voucher specimen was deposited (collection number AR010) for future reference. 

### 4.2. Chemicals and Drugs 

Analytical TLC was performed on a pre-coated aluminium backed silica gel 60 F_254_ plates (Merck KGaA, Darmstadt, Germany). Chloroform and methanol (HPLC grade, Carlo Erba, Milan, Italy). Dimethylsulfoxide (DMSO) (Sigma-Aldrich Co., St. Louis, MO, USA), amphotericin B (Laborchemikalien GmbH, Seelze, Germany) and Diminasan^®^ (1.05 g diminazene diaceturate × 1.31 gantipyrine (Woerden, the Netherlands) were used in the biological assay.

### 4.3. Instruments 

A Rotary evaporator R-200 (Buchi, Switzerland) was used to remove organic solvents. TLC chromatograms were viewed under a UV cabinet (CAMAG, Muttenz, Switzerland), ^1^H-NMR and ^13^C-NMR spectra were recorded at a room temperature on a Bruker Avance DMX400 FT-NMR spectrometer (Bruker, Billerica, MA, USA). MS was run on an ultra-high-performance liquid chromatography-mass spectrometer (UHPLC-MS) Waters^®^ Acquity UHPLC system (Waters, Milford, MA, USA) equipped with electrospray ionization (ESI) in the negative mode.

### 4.4. Experimental Animals 

Swiss albino mice (20–35 g) of either sex were obtained from the School of Pharmacy breeding colony, AAU. The animals were kept at room temperature and 12 h light and dark cycle in polypropylene cages, and provided with commercial pelleted ration and clean water ad libitum. The animals were acclimatized for one week under controlled conditions before conducting the experiments. All the experiments were conducted in accordance with internationally accepted laboratory animal use and care guideline [44] and were approved by the Institutional Review Board of School of Pharmacy, AAU (approval code: ERB/SoP/028/01/21).

### 4.5. Leaf Latex Preparation 

Latex was collected from the leaves of *A*. *rugosifolia* by cutting the leaves at the bottom and allowing the latex to drain on a plastic sheet. The latex was then left to dry in air for 3 days, which yielded a light powder, then the dried material was weighed and stored in a refrigerator at 4 °C until use.

### 4.6. Isolation of Plicataloside 

The latex was dissolved in methanol, then filtered and dried under reduced pressure using A Rotary Evaporator R-200 (Buchi, Switzerland). The dried extract was subjected to PTLC over silica gel plates using a mixture of chloroform and methanol (2:1) as a mobile phase. Chromatographic zones were visualized under UV light of 254 nm and 366 nm.

### 4.7. Spectral Data Plicataloside

Plicataloside was isolated as a light-brown-coloured amorphous substance; *R_f_* = 0.27 (CHCl_3_/MeOH; 2:1); HRESI-MS (-ve mode, Appendix A) *m/z*: 513.1622 [M−H]^−^ (exact calcd. 513.1608 for [M−H]^−^) indicating a molecular formula of C_23_H_30_O_13_; *m/z*: 351.1095 [M−Glc−H]^−^; 189.0565 [M−Glc−Glc−H]^−^; ^1^H-NMR (400 MHz, DMSO-d_6_, Appendix A) δ: 2.38 (3H, *s*, Me), 2.51-3.44 (12H, *m*, H-2′-H-6′ & H-2″-H-6″), 4.82 (1H, *d*, J = 8 Hz, H-1′), 4.99 (1H, *d*, J = 8 Hz, H-1″), 7.19 (1H, *brs*, H-4), 7.19 (1H, *brd*, J = 8 Hz, H-7), 7.26 (1H, *t,* J = 8 Hz, H-6), 7.40 (1H, *brd,* J = 8 Hz, H-5), 9.30 (1H, *s*, OH-1); ^13^C-NMR & DEPT (100 MHz, DMSO-d_6_, Appendix A) δ: 18.2 (C-11), 61.18 (C-6′), 61.32 (C-6″), 70.29 (C-4′ & C-4″), 74.05 (C-2″), 74.66 (C-2′), 76.59 (C-3″), 76.75 (C-3′), 77.47 (C-5′), 77.92 (C-5″), 103.05 (C-1″), 104.58 (C-1′), 110.25 (C-7), 115.21 (C-9), 119.45 (C-4), 122.50 (C-5), 125.85 (C-6), 133.25 (C-10), 134.00 (C-3), 140.04 (C-2) 144.49 (C-1) 154.37 (C-8).

### 4.8. Acute Toxicity Study 

Acute oral toxicity of plicataloside was carried out according to the OECD guidelines for testing of chemicals on Swiss albino mice [24]. Ten healthy Swiss female mice weighing 23–25 g was randomly divided into 2 groups of 5 mice per group. After fasting for 3 h, mice in the first group were given 2 g/kg of plicataloside and the second group received 0.2 mL distilled water (control group) orally and observed for any signs of toxicity for 14 days to assess safety of the test substances. The mice were observed for any gross behavioral changes (loss of appetite, hair erection, lacrimation, mortality) and other signs of toxicity manifestation.

### 4.9. Isolation of Trypanosoma congolense 

*Trypanosoma congolense* field isolate was collected from Oromia region, Buno Bedele Zone, Dabu Woreda, Sebategna kebele, southwestern Ethiopia. The Murray method was used to check the presence of *T. congolense* in cattle [45]. The presence of *T. congolense* parasites was identified based on their type of motility and further confirmed by the National Tsetse and Trypanosomiasis Control Centre (NTTCC) in Bedelle town by preparing thin blood smear. The slides were read using an oil immersion 100x objective after fixing with methanol and staining with Giemsa stain [46,47]. The stained field was then compared with that of the reference species. This was followed by collection of blood from the positive cattle jugular vein using heparinized vacutainer tube. Then, 0.2 mL of blood containing the parasites was injected intraperitoneally to 10 healthy laboratory mice [32], and transported to the Department of Pharmaceutical Chemistry and Pharmacognosy, College of Health Sciences, AAU, Addis Ababa for serial passage to other mice used in the experiments.

### 4.10. Leishmania Test Strains 

In vitro antileishmanial activity tests were performed on the promastigote stage of *L. aethiopica* and *L. donovani* clinical isolates obtained from Leishmania Diagnostic and Research Laboratory collection, Faculty of Medicine, AAU. They were grown in tissue culture flasks containing RPMI 1640 medium supplemented with 10% heat-inactivated fetal calf serum, 100 IU penicillin/mL, and 100 μg/mL streptomycin solution at 24 °C for *L. donovani* and 22 °C for *L. aethiopica.* In vitro cell-free medium was used to set up the test system for the determination of IC_50_ values of the test substances and to grow the parasites [36,48].

### 4.11. In Vitro Antitrypanosomal Activity Tests 

Assessment of in vitro antitrypanosomal activity of the test substances was performed in triplicates in 96-well microtiter plates. Infected mice with a high parasitemic state were sacrificed and blood was collected in an ethylenediaminetetraacetic acid (EDTA; Thermo Fisher Scientific, Waltham, MA, USA) tube prepared with phosphate buffer saline glucose (PBSG; Gibco, Waltham, MA, USA). A measure of 20 μL of blood containing about 20–25 parasites per field was mixed with 5 μL of each of the test substances dissolved in 10% DMSO at concentrations of 20.0, 10.0, 2.0, and 0.5 mg/mL to produce 4.0, 2.0, 0.4, and 0.1 mg/mL of final concentrations, respectively [49]. A set of negative controls containing the parasite (20 μL of infected blood) suspended in 10% DMSO was introduced to ensure that the effect monitored was that of the test substances alone. Tests were also performed with similar concentrations of Diminasan^®^ to serve as a positive control [30,50]. 

After 5 min of incubation in closed microtiter plates maintained at 37 °C, the parasites were observed every 5 min for death/motility using 400× objective for a total of 60 min by placing separately 2 μL of test mixtures on microscope slide and covered with 22 mm × 22 mm cover slips [51]. 

### 4.12. In Vivo Infectivity Test 

An in vivo infectivity test was performed to establish whether there were any remaining infective parasites after the in vitro test. In this test, the inoculum contained in the remaining incubation mixture from each well of the micro titer plate where the in vitro test was carried out, i.e., 4, 2, 0.4, and 0.1 mg/mL of test substance separately in 0.02 mL of infected blood as described elsewhere [25]. A total of 65 healthy mice (5 animals per dose) received each mixture intraperitoneally (after 2 h of incubation period) and were observed for 30 days for the development of infection.

### 4.13. In Vitro Antileishmanial Activity Assay 

To a separate 96-well microtiter plate containing 100 µL of complete culture medium, each of the test substances was added to triplicate wells to achieve a final concentration of 100 µg/mL. Then, 100 µL suspension of the parasites containing 3.5 × 10^6^ promastigotes of *L. aethiopica* or *L. donovani* were added to each well, which was obtained from the previous culture. After 68 h of incubation, 20 µL (10% of the total volume of each well) resazurin (0.125 mg/mL) was added and covered with aluminum foil, and left at room temperature. After a total incubation period of 72 h, fluorescence intensity was measured using Victor3 Multilabel Counter (PerkinElmer, Waltham, MA, USA) at an emission wavelength of 590 nm and excitation wavelength of 544 nm. During the assay, cell viability was monitored by measuring the fluorescent signal. The number of viable cells was proportional to the fluorescence intensities produced [36,48].

### 4.14. Data Analysis

Antileishmanial activity (IC_50_) values were expressed as mean ± SD of triplicate experiments, and they were evaluated from sigmoidal dose-response curves of percent inhibition using GraphPad Prism 8 (GraphPad Software, CA, USA). Values of antitrypanosomal activity were expressed as mean ± SEM using Microsoft Excel 2013.

## 5. Conclusions

The results of the current study revealed that the leaf latex of *A. rugosifolia* possesses genuine in vitro antitrypanosomal activity against field isolate of *T. congolense* and antileishmanial effect against *L. aethopica* and *L. donovani* promastigotes. Phytochemical investigation of the latex afforded a rare naphthalene derivative identified as plicataloside. Plicataloside showed superior activity against these parasites when compared to the effect of the latex. It can therefore be concluded that the antitrypanosomal and antileishmanial effects of the latex of *A. rugosifolia* are attributed in full or in part to the presence of plicataloside. Furthermore, plicataloside has the potential to be used as a lead compound for the development of cost-effective and more potent alternative drugs for the treatment of trypanosomiasis and leishmaniasis.

## Figures and Tables

**Figure 1 molecules-27-01400-f001:**
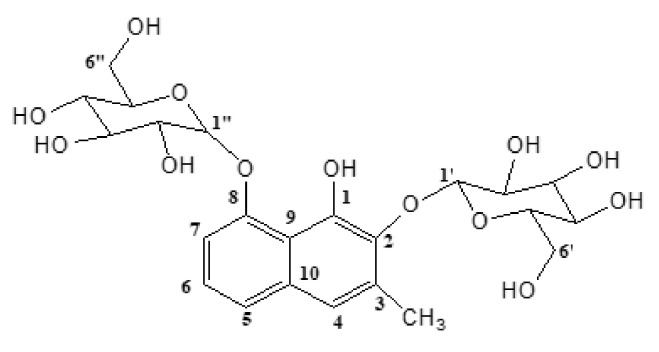
Structural formula of plicataloside.

**Table 1 molecules-27-01400-t001:** Effect of the latex of *Aloe rugosifolia* and plicataloside on motility of *Trypanosoma congolense*.

	Time (min) After Which Motility Ceased			
Concentration (mg/mL)	**Latex**	Plicataloside	Diminazene Diaceturate	10% DMSO
4.0	35	30	20	NE
2.0	50	45	35	NE
0.4	>60	60	50	NE
0.1	>60	>60	>60	NE

Note: DMSO: Dimethyl sulfoxide, NE: No noticeable effect on motility even after 60 min.

**Table 2 molecules-27-01400-t002:** Duration (days) after which parasitaemia developed in mice inoculated with a mixture of test substances and *Trypanosoma congolense* infected blood.

Dose of Test Substance (mg/mL) Mixed with 0.02 mL of Infected Blood		Number of Mice Which Developed Infection	Infection Interval in Days (mean ± SEM)
Latex	4.0	1/5	24.00
	2.0	3/5	14.66 ± 0. 66
	0.4	5/5	13.00 ± 0.55
	0.1	5/5	12.4 ± 0.60
Plicataloside	4.0	0/5	Ni
	2.0	0/5	Ni
	0.4	3/5	18.33 ± 0.66
	0.1	5/5	14.20 ± 0.49
Diminazene Diaceturate	4.0	0/5	Ni
	2.0	0/5	Ni
	0.4	2/5	19.50 ± 0.50
	0.1	5/5	15.20 ± 0.37
Dimethylsulfoxide	0.1 mL	5/5	11.60 ± 0.24

Note: Ni: no infection developed in the observation period, SEM = Standard error of mean.

**Table 3 molecules-27-01400-t003:** Antipromastigote activity of latex and plicataloside isolated from the leaf latex of *Aloe rugosifolia* against *Leishmania aethiopica* and *Leishmania donovani*.

Test Samples/Drugs	IC_50_; µg/mL (µM)	IC_50_; µg/mL(µM)
	Against *L. aethiopica*	Against *L. donovani*
Latex	24.50 ± 0.24	31.21 ± 0.01
Plicataloside	14.22 ± 0.41(27.66 ± 0.80)	18.86 ± 0.03 (36.69 ± 0.06)
Amphotericin B	8.10 ± 0.11 (8.77 ± 0.12)	7.20 ± 0.15 (7.79 ± 0.16)
Media alone (NC)	0.00	0.00
1% Dimethyl sulfoxide (NC)	0.00	0.00

Values expressed as mean ± SD (n = 2); NC: Negative control; IC_50_: Effective concentration required to achieve 50% growth inhibition in μg/mL (μM).

## Data Availability

The authors declare that all data supporting the finding of this study are included in this article.

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
