# Peer review of "In Vitro Antileishmanial and Antitrypanosomal Activities of Plicataloside Isolated from the Leaf Latex of Aloe rugosifolia Gilbert & Sebsebe (Asphodelaceae)"

_molecules, 2022, doi:10.3390/molecules27041400_

Round 1

Reviewer 1 Report

Reviewer comments (Manuscript ID: molecules-1586235)

Molecules Editorial Office

The manuscript "In Vitro Antileishmanial and Antitrypanosomal Activities of Plicataloside Isolated from the Leaf Latex of Aloe rugosifolia Gilbert & Sebsebe" describes the in vitro evaluation of antitrypanosomal activity of the leaf latex of A. rugosifolia against Trypanosoma congolense and also, its antileishmanial activity against the promastigotes of Leishmania aethiopica and Leishmania donovani. Moreover, the authors have undertaken the isolation of the major constituent of this plant by preparative TLC. The manuscript is an important contribution in the terms of selective isolation and the antiparasitic activities of the plant and its main constituent. However, there were some deductions that need to be reviewed carefully. Please review the corrections done for your manuscript.

In the Title, please In Vitro in italic and you can also add the plant’s family after the botanic authority.

In abstract,

Page 1, lines 14, 16, in vitro and in vivo in italic. Please, correct it in all the text,

In Keywords, add the plant’s family.

Aloe rugosifolia; Trypanosoma congolense;, Leishmania aethiopica; Leishmania donovani in italic.

In Main text

Page 2, lines 72 and 73, instead “Diglucosylation was evident from the 13C-NMR spectral data (two anomeric carbons (δ 103.05, 104.58), two sp3 oxymethylene carbons (δ 61.18, δ 61.32)” of is better to write “Diglucosylation was evident from the 13C-NMR spectral data (two anomeric carbons (δ 103.05, 104.58), two sp3 oxymethylene carbons (δ 61.18, 61.32)” , to uniform.

Page 2, line 78, instead of δ2.51-3.44 ppm, and please, write δ 2.51-3.44. You cannot use at the same time δ and ppm.

In the text, your description does not imply really, the configuration of those sugars and also their positions at C-2 and C-8. You can add the couple constant to justify their configuration. We did not convince that the configuration is β.

Why you do not think that these sugars could be at ortho position (C-1 and C-2) or at C-1 and C-8? Even the position of the methyl at position C-3 is not clear. The description is very simple. This compound described by authors is a known one but they can add a 2D-NMR as HMBC to justify the position of each substituent on naphthalene skeleton. Please add this supplementary spectrum to convince the readers.

Also, can you write the aromatic protons with their multiplicities and couple constant, in the main text and in spectral data part?

The biological activities of this study are relevant but why you didn’t undertake the total study of the plant to isolate several natural products to enhance your work?

In Materials and Methods

Spectral Data Plicataloside:

Page 6, line 245, please write (C-4' & C-4"),

Page 8, line 324, instead of “Neative-mode HR-ESI-mass spectrum of plicataloside” is better to write “Negative-mode HR-ESI-mass spectrum of plicataloside”

References

Page 9, line 365, please, abbreviate “Scientific World Journal”

Page 9, line 367, please, abbreviate “Weekly Epidemiological Record

In vitro and in vivo in italic in all the references.

Author Response

Point-by-point Response to the Reviewer’s comments

Ref.: Ms. No. molecules-1586235
Dear Editor,

Thank you very much for the review of our manuscript entitled: “In Vitro Antileishmanial and Antitrypanosomal Activities of Plicataloside Isolated from the Leaf Latex of Aloe rugosifolia Gilbert & Sebsebe”. We sincerely appreciate all valuable comments and suggestions, which helped us to improve the quality of the article. Our responses to the Reviewer’s comments are described below in a point-by-point manner.

Reviewer Report-1

  1. In the Title,

(1a) please In Vitro in italic

  • Molecules do not italicize in vitro and in vivo. That is why we left it unitalicized.

(1b)  you can also add the plant’s family after the botanic authority

  • Done
  1. In abstract,

Page 1, lines 14, 16, in vitro and in vivo in italic. Please, correct it in all the text,

  • Please see 1a
  1. InKeywords,

(a) add the plant’s family.

  • Done

(b) Aloe rugosifolia; Trypanosoma congolense;, Leishmania aethiopica; Leishmania donovani in italic.

  • Done
  1. In Main text

(4a). Page 2, lines 72 and 73, instead “Diglucosylation was evident from the 13C-NMR spectral data (two anomeric carbons (δ 103.05, 104.58), two sp3 oxymethylene carbons (δ 61.18, δ 61.32)” of is better to write “Diglucosylation was evident from the 13C-NMR spectral data (two anomeric carbons (δ 103.05, 104.58), two sp3 oxymethylene carbons (δ 61.18, 61.32)” , to uniform.

  • Corrected
  •  
  • (4b). Page 2, line 78, instead of δ2.51-3.44 ppm, and please, write δ 2.51-3.44. You cannot use at the same time δ and ppm.

Corrected

  • (4c). In the text,your description does not imply really, the configuration of those sugars and also their positions at C-2 and C-8. You can add the couple constant to justify their configuration. We did not convince that the configuration is β.
  •  
  • β-D-glucose was deduced from the large coupling constant (8 Hz) of its anomeric proton

(4d). Why you do not think that these sugars could be at ortho position (C-1 and C-2) or at C-1 and C-8? Even the position of the methyl at position C-3 is not clear. The description is very simple. This compound described by authors is a known one but they can add a 2D-NMR as HMBC to justify the position of each substituent on naphthalene skeleton. Please add this supplementary spectrum to convince the readers.

  • The positions of the sugar units was confirmed by HMBC experiments. The following new sentence has been added in the main text (Page 2; Lines 80-83). “The positions of attachmentof the two O-glucosyl units at C-2 and C-8 were confirmed by the HMBC experiments, which revealed correlations between the anomeric proton H-1' (δ 4.82, d) and C-2 (δC 140.04) and also between the anomeric proton H-1" (δ 4.99, d) and C-8 (δC 154.37)”

 (4e). Also, can you write the aromatic protons with their multiplicities and couple constant, in the main text and in spectral data part?

  • Done

(4f). The biological activities of this study are relevant but why you didn’t undertake the total study of the plant to isolate several natural products to enhance your work?

  • The latex of the plant contains only two compounds (one major and the other was a very minor compound.
  1. In Materials and Methods

(5b). Spectral Data Plicataloside: Page 6, line 245, please write (C-4' & C-4"),

  • Corrected

(5c). Page 8, line 324, instead of “Negative-mode HR-ESI-mass spectrum of plicataloside” is better to write “Negative-mode HR-ESI-mass spectrum of plicataloside”

  • Corrected
  1. References

(6a). Page 9, line 365, please, abbreviate “Scientific World Journal”

  • Scientific World Journal is now abbreviated as World J.

(6b). Page 9, line 367, please, abbreviate “Weekly Epidemiological Record

  • ‘Weekly Epidemiological Record is now abbreviated as ‘ Epidemiol. Rec

(6c). In vitro and in vivo in italic in all the references.

  • Please see 1a.

Reviewer 2 Report

In the present manuscript, the authors give a detailed characterization of antitrypanosomal and antileishmanial activities of a novel compound, plicataloside, isolated from aloe latex. Overall, the studies were well performed and clearly merit attention. However, a few details could be improved. In order to compare the efficacies of plicataloside with established drugs such as amphotericin B or Diminazene Di-aceturate, it would be helpful to give inhibitory concentrations not only on a weight basis (mg/l), but also on a molar basis (mM). The discussion contains repetitions of the results and could therefore be shortened.

Author Response

Point-by-point Response to the Reviewer’s comments

Ref.: Ms. No. molecules-1586235

Dear Editor,

Thank you very much for the review of our manuscript entitled: “In Vitro Antileishmanial and Antitrypanosomal Activities of Plicataloside Isolated from the Leaf Latex of Aloe rugosifolia Gilbert & Sebsebe”. We sincerely appreciate all valuable comments and suggestions, which helped us to improve the quality of the article. Our responses to the Reviewer’s comments are described below in a point-by-point manner.

 Reviewer Report-2

(1). However, a few details could be improved. In order to compare the efficacies of plicataloside with established drugs such as amphotericin B or Diminazene Di-aceturate, it would be helpful to give inhibitory concentrations not only on a weight basis (mg/l), but also on a molar basis (mM).

  • All IC50 values are now given in µM also.

(2). The discussion contains repetitions of the results and could therefore be shortened.

  • Done

Reviewer 3 Report

  1. The symptoms, complications of trypanosomiasis and leishmaniasis diseases and the consequences on society and the economy should be mentioned.
  2. Plicataloside as active compound has any toxicity?
  3. Line 47-49: “They have become more difficult to control due to a number of factors that limit the utility of current drugs in resource poor settings”, explain.
  4. The anti-protozoal mechanism of action of A. rugosifolia and its active constituents should be included.
  5. Extraction proceduces should be added.
  6. The authors should mention the source of all chemicals and cell lines used throughout the manuscript like EDTA, PBSG, …etc.
  7. The manuscript contains typo errors such as Aloe rugosifolia, in vivtro, in vivo,…etc
  8. The graphical abstract is highly recommended.
  9. The authors could benefit from the following reference: “ Saeed, A.,et al., . (2018). Synthesis, Antibacterial and Antileishmanial Activity, Cytotoxicity, and Molecular Docking of New Heteroleptic Copper (I) Complexes with Thiourea Ligands and Triphenylphosphine. Russian Journal of General Chemistry, 88(3).‏

Author Response

Comments and Suggestions for Authors

  1. The symptoms, complications of trypanosomiasis and leishmaniasis diseases and the consequences on society and the economy should be mentioned.
  • Done
  1. Plicataloside as active compound has any toxicity?
  • We have now added the acute toxicity result of plicataloside.

3. Line 47-49: “They have become more difficult to control due to a number of factors that limit the utility of current drugs in resource poor settings”, explain.

Two new references have been added to show the current drugs limitations.

4. The anti-protozoal mechanism of action of A. rugosifolia and its active constituents should be included.

We have not studied the mechanism of action of the isolated substances

5. Extraction procedures should be added.

Added

6. The authors should mention the source of all chemicals and cell lines used throughout the manuscript like EDTA, PBSG, …etc.

Mentioned

7. The manuscript contains typo errors such as Aloe rugosifolia, in vivtro, in vivo,…etc

Corrected

8. The graphical abstract is highly recommended.

Molecules do not require graphical abstract.

The authors could benefit from the following reference: “Saeed, A.,et al., (2018). Synthesis, Antibacterial and Antileishmanial Activity, Cytotoxicity, and Molecular Docking of New Heteroleptic Copper (I) Complexes with Thiourea Ligands and Triphenylphosphine. Russian Journal of General Chemistry, 88(3).

Reviewer 4 Report

the overall research in applicable from medical and pharmaceutical point of view. So I recommend the publication of this piece of work although the compound is not new. 

Author Response

No comments or suggestions were given by this reviewer.

Round 2

Reviewer 1 Report

The authors must correct recommended remarks.

Author Response

We have uploaded the revised version of our manuscript coded MS no.:  molecules-1586235 as per the comments of the editor and reviewers. For the sake of your quick scrutiny, all changes that have been made are marked using Track Changes system. Please refer below to a point-by-point response to the reviewer comments

----------------------------

Response to Reviewer comments (Manuscript ID: molecules-1586235)

 1. In Main text

Page 2, lines 94 and 95, instead “between the anomeric proton H-1' (δ 4.82, d) and C-2 (δC 140.04) and also between the anomeric proton H-1" (δ 4.99, d) and C-8 (δC 154.37).” of is better to write “between the anomeric proton H-1' (δ 4.82, d) and C-2 (δ 140.04) and also between the anomeric proton H-1" (δ 4.99, d) and C-8 (δ 154.37)” , to uniform (see line 86).

  • Done

2. In all the text, in vitro and in vivo in italic, even in the references part. Please correct

  • Done
